# Microstructure Evolution and Mechanical Properties of Needle-like ZrB_2_ Reinforced Cu Composites Manufactured by Laser Direct Energy Deposition

**DOI:** 10.3390/mi13020212

**Published:** 2022-01-28

**Authors:** Xiangzhe Lv, Zaiji Zhan, Haiyao Cao

**Affiliations:** State Key Laboratory of Metastable Materials Science & Technology, Yanshan University, Qinhuangdao 066004, China; lvxiangzhe@hotmail.com (X.L.); haiyaocao@ysu.edu.cn (H.C.)

**Keywords:** laser direct energy deposition, ZrB_2_, copper matrix composite, microstructure

## Abstract

Laser additive manufacturing is an advanced material preparation technology, which has been widely used to prepare various materials, such as polymers, metals, ceramics and composites. Zirconium diboride (ZrB_2_) reinforced copper composite material was fabricated using laser direct energy deposition technology. The microstructure and phase composition of the composite material were analyzed, and the influence of laser energy density on the microstructure and mechanical properties of composite materials was discussed. The results showed that the needle-like ZrB_2_ ceramic reinforcement was successfully synthesized via an in-situ synthesis reaction. The composites were mainly composed of needle-like ZrB_2_, Ni dendrites and a Cu matrix. The morphological changes of Ni dendrites could be observed at the interface inside the composite material: cellular crystals → large-sized columnar dendrites → small-sized dendrites (along the solidification direction). The continuous Ni dendritic network connected the ZrB_2_ reinforcements together, which significantly improved the mechanical properties of the composite material. At a laser energy density of 0.20 kJ/mm^2^, the average microhardness of the composite material reached 294 HV_0.2_ and the highest tensile strength was 535 MPa. With the laser energy density increased to 0.27 kJ/mm^2^, the hardness and tensile strength decreased and the elongation of the Cu composites increased due to an increase in the size of the ZrB_2_ and a decrease in the continuity of the Ni dendritic.

## 1. Introduction

Laser additive manufacturing (LAM) is an advanced material preparation technology, which adopts laser as an energy source and powders or wires as raw material to construct a 3D specimen under the control of a computer [1,2]. LAM has been widely used to prepare various materials, such as polymers, metals, ceramics and composites, and has been applied in aviation, medicine, transportation, machinery and other fields [3,4,5]. The requirements for material molding, the uniform dispersion of reinforcements and the formation of a strong interfaces can be met simultaneously by LAM [6,7]. LAM can be divided into two types according to the feeding methods: powder bed LAM and powder feeder LAM [8]. Laser direct energy deposition (LDED) is a LAM technique that uses the powder feeder [9,10]. During its processing, the laser beam melts the powder and forms a molten pool to achieve the manufacture, surface repair and surface modification of the specimen [11,12]. For example, high-strength TiC-NiTi composites are manufactured using LDED [13]; functional gradient material of Ti6Al4V/316L with different proportions has been successfully fabricated using laser melting deposition [14]. Furthermore, damaged railway wheels have been repaired by depositing a stainless-steel layer on the wheel surface using LDED technology [15].

Copper is a common and important metal material due to its excellent thermal conductivity, corrosion resistance and ductility [16,17], and it had been widely used in electronics, transportation, aviation, the military and other fields [18,19]. However, with the rapid development of industry and the transformation of service environments, high requirements have been put forward for materials in terms of offering comprehensive properties. Due to its low strength and poor wear resistance, pure copper material does not meet such requirements. The compounding of multiple element materials has been one of the significant trends in the development of new materials [20]. Ceramic materials possess high hardness and strength, outstanding wear resistance and high temperature stability, and have become important reinforcement materials in composites [21,22]. For example, ZrB_2_/Cu_5_Zr reinforced Cu composites have exhibited a good combination of electrical conductivity and mechanical properties [23]. ZrB_2_ ceramic whisker is a single crystal with a very small diameter [24]. Its atomic arrangement is highly ordered without defects, so its strength is potentially close to the crystal’s theoretical strength. Therefore, a ceramic whisker reinforcement can be added to the copper matrix to improve the mechanical properties of copper, increase its service life and extend its scope of application.

With regard to ceramic reinforced metal matrix composites (CMC), the manufacture of defect-free CMCs with excellent mechanical properties has always presented a challenge and has been the focus of research due to large differences in the physical and chemical properties of ceramic reinforcements and metal matrices [1,25]. According to the addition method, the reinforcements of CMCs can be divided into external ceramic and in-situ synthesized ceramic reinforcements [26]. In-situ synthesized ceramic reinforcements have shown good compatibility and a strong interface with the metal matrix [27]. Therefore, the reinforcement efficiency of in-situ synthesized ceramic can be maximized [28]. Furthermore, the laser additive manufacturing of CMCs has an advantage in that the molten pool environment formed in the laser processing process can provide the basic conditions for the in-situ synthesis of ceramic reinforcements. For example, TiB and TiC was synthesized in a liquid Ti melt pool successively via the direct laser deposition of a mixture of Ti and B_4_C, and they acted as the heterogeneous nucleation sites for the Ti matrix, which promoted the formation of fine equiaxed α-Ti grains [29]. In the process of depositing Ta-reinforced NiCrBSi coating on the surface of 35 CrMo steel substrate, a variety of reinforcement phases (TaC, Ni_3_Fe, CrB, M_23_C_6_ and Cr_2.8_Fe_1.2_B_4_) were synthesized in the molten pool, which significantly improved the high temperature wear resistance of the coating [30]. During the laser sintering of Cu-4.1Zr-1.1B powder, ZrB_2_ and CuZr were synthesized in the Cu matrix, which improved the high temperature hardness of Cu composites [31].

In this study, the exploration goal was to use LDED technology to prepare ZrB_2_ ceramic whisker reinforced copper matrix composites. The ZrB_2_ reinforcement was in-situ synthesized via the designed chemical reaction in the molten pool, and the crystal structure of its growth tip was analyzed. The microstructure, phase composition and interface characteristics of the composites were determined. The influence of energy input density on the microstructure of composites was discussed. In addition, the microhardness distribution and tensile properties of the samples prepared under different energy input densities were studied.

## 2. Materials and Methods

The composite powders ZrO_2_, Al, Ni-B_4_C (B_4_C, 60 wt%), Ni and Cu were used as the raw material for laser direct energy deposition. The sizes and weight percentages of the composite powders were listed in Table 1. ZrO_2_, Al and B_4_C were the reactants of the in-situ synthesis reaction, and the molar ratio between them was set from the designed reaction:(1)$$4\mathrm{Al}+3{\mathrm{ZrO}}_{2}+B_{4}C\;\to \;2{\mathrm{Al}}_{2}O_{3}+2{\mathrm{ZrB}}_{2}+\mathrm{ZrC}\;\Delta H_{298}^{0}=-829kJ\;\Delta G_{298}^{0}=-807kJ$$

The Gibbs free energy (ΔG) of Equation (1) was negative, indicating that the reaction could proceed spontaneously. Previous studies have also confirmed the successful synthesis of in-situ ZrB_2_ ceramic reinforcements using this ratio of reactants [32]. The raw powders were mixed in a V series mixer at a rotation speed of 20 r/min for 120 min to prepare the composite powders. The mixtures were dried in a vacuum drying oven (150 °C, 1 h). Ni was added to improve wettability and the bond strength between the ceramic and the Cu matrix [27,33]. In addition, Ni was also beneficial to the stability of the molten pool and the surface morphology of the deposited layer. Figure 1 showed the morphology of the composite mixed powders.

The detailed experimental process is shown in Figure 2a. The laser direct energy deposition system was composed of a semiconductor fiber laser (ZKSK-3008), mechanical hand (YASKAWA, MH24), synchronous powder feeder (DPSF-2), workbench and computer. The wavelength of the laser beam was 1024 nm, and the distance from the laser processing head to the surface to be processed was 180 mm. The composite powders were supplied to the laser molten pool via coaxial powder feeding, and Ar was used as the carrier gas and protective gas. Five layers were deposited using the cyclic reciprocating scanning strategy. As shown in Figure 2b, the reference axis of the laser direct energy deposition system was defined in terms of the following: the building direction (BD), the scanning direction (SD) and the lateral direction (LD).

Pure copper with a size of 50 × 50 × 15 mm^3^ was used as the substrate for laser direct energy deposition. The copper substrate was polished using sandpaper and cleaned with alcohol prior to laser processing. The process parameters of laser direct energy deposition were set according to the pre-experiment: an overlap rate of 50%, a powder feeding rate of 2 g/min, a scanning speed (V) of 180 mm/min, a spot diameter (D) of 3 mm and a laser power (P) of 1.8 or 2.4 kW. The laser energy density (E = P/VD) represented the laser energy applied per unit area during the deposition process, which was used to comprehensively analyze the influence of process parameters for each sample. Two samples were prepared under different laser energy densities; the E of samples 1 and 2 were 0.20 and 0.27 kJ/mm^2^, respectively.

The samples were cut using wire electric discharge machining, then sanded, polished and corroded. The corrosive was composed of FeCl_3_ (10 g), HCl (37%, 10 mL) and H_2_O (100 mL). The phase composition of the composite material was analyzed using an X-ray diffractometer (XRD, D/Max-2500PC) at a speed of 0.5°/min. A tungsten wire scanning microscope (SEM, KYKY3200) equipped with an energy spectrometer (EDS) was used to observe the microstructure of the composites and the morphology of the reinforcements. The growth direction and interface characteristics of the reinforcement were characterized using transmission electron microscopy (TEM, FEI’s Tecnai G2 F30 S-TWIN). A disc with a diameter of 3 mm was cut on the SD-LD surface of the composites, and then a TEM sample was obtained via mechanical thinning and ion beam thinning. The microhardness distribution of the samples on the BD-LD was tested using a Vicker’s hardness tester (HVS-1000) under a load of 0.2 kg and the dwell time of 10 s; the distance between the test points was 100 μm. The tensile samples of the composites were cut on the SD-LD plane (the specific location is shown in Figure 2b and the size of the tensile samples is given in Figure 2c). The mechanical properties of the composites were tested at ambient temperature using a universal tensile testing machine (TH5000) with a testing speed of 0.3 mm/min. Three replicas were tested under the same test conditions. In addition, the fracture morphology of the composites was characterized using SEM.

## 3. Results

### 3.1. Macroscopic Morphology

Figure 3a shows the macroscopic morphology of sample 1 prepared using LDED. The size of the sample was approximately 40 mm × 40 mm × 5 mm, and its surface was flat and clean with a light golden color. Track interfaces could be observed on the surface, which were formed by the overlapping of the deposition track in parallel to the SD. The cross-section image of the sample at the remarked position in Figure 2a is shown in Figure 3b. No defects such as pores, cracks and impurities were seen on the surface or at the interface.

### 3.2. Phase Constituents

The XRD patterns of the Cu composites prepared at different laser energy densities are shown in Figure 4. The phase constituents of the composites were Cu, ZrB_2_, Ni, ZrC and Al_2_O_3_. Due to the internal stress caused by rapid solidification and the mutual dissolution between Ni with Cu, the peaks of Cu and Ni were shifted slightly. The phase constituents from the in-situ chemical reaction were exactly as expected, namely the target reinforcements were successfully synthesized in the Cu matrix. Two samples prepared under different laser energy densities showed the same phase composition, indicating that the further products of the in-situ synthesis reaction would not be affected by the laser energy density within a suitable range.

### 3.3. Microstructures of the Composites

The microstructures of ZrB_2_ reinforced Cu composites were analyzed using SEM. Figure 5a shows the SEM image of sample 1 on the BD-LD plane. It could be seen that the track interfaces (white dotted line) and layer interfaces (yellow dotted line) formed during deposition and overlapping. The track interface is the molten pool boundary during the cyclic overlapping in a deposition layer. The layer interface is the interface between deposition layers. There were no defects such as holes, impurities and cracks in the deposition track (molten pool) and the interfaces, indicating that a dense and defect-free ceramic reinforced composite material can be prepared by using LDED with a suitable overlap rate and laser energy density. Figure 5b shows an enlarged SEM image of the middle (dotted box 1) of the molten pool. The needle-like phase and dendrites were evenly distributed in the matrix. And some secondary dendrite arms were observed parallel to each other. According to the XRD result (Figure 4) and a previous study [30], it could be determined that the dendrite is Ni solid solution and the matrix was Cu. The microstructure of the track interface (dotted box 2) is shown in Figure 5c. A small amount of needle-like phase was observed at the interface. There was a large-sized dendrite region with a width of about 50 μm at the interface, and the large-sized dendrites were directly combined with the dendrites on both sides of the interface. Figure 5d shows the microstructure of the bottom (dotted box 3) of the molten pool (near the layer interface). It could be observed that the columnar dendrites were arranged parallel to the BD. At the bottom of the molten pool formed by laser irradiation, the heat of the molten pool was mainly conducted to the previous deposition layer, and the main heat flow (temperature gradient) direction was parallel to the BD [34], so vertical columnar dendrites were formed. The SEM image of the layer interface (dotted box 4) is shown in Figure 5e. A large columnar dendrite region with a width of about 55 μm on the interface could be observed. Compared with the dendrites on both sides of the interface, the number of secondary dendrites decreased significantly in the large dendrite region. There was a transition zone (large dendritic region) at the track interface and the layer interface, which helped to reduce the crack and stress concentration and improve the bonding strength.

During the LDED, the size and lifetime of the molten pool increased relatively with the increase in energy input, so the heat of the molten pool transferred to the previous deposition layer increased and the area of the remelting and heat-affected zones of the laser processing enlarged, which had an impact on the microstructure of the composites. Figure 6a shows the microstructure of sample 2 on the BD-LD plane at a high laser energy density of 0.27 kJ/mm^2^. The evolution of the crystal morphology could be clearly observed. The position marked by the white dashed line is the remelted interface (track interface). According to the build direction, the left side of the interface was N + 1 track deposition, in which small-sized dendrites and large-sized columnar dendrites could be observed. Compared with the area in sample 1 (Figure 5c), the area containing large-size columnar dendritic was increased. The right side of the interface was N track deposition, and its microstructure was mainly cellular crystals. At the laser remelting interface (track interface), the liquid phase was put in direct contact with the solid phase of a similar composition in order to achieve epitaxial growth, and its solidification process did not involve nucleation. Figure 6b shows the enlarged SEM image of the track interface in Figure 6a. Along the yellow dashed line, from the bottom right to the top left, the morphologies of cellular crystals, large-sized columnar dendrites, and small-sized dendrites are recorded. The evolution of the microstructure was closely related to changes in the thermodynamic conditions. The morphology of the dendrites was related to the ratio of temperature gradient (G) to solidification rate (R), which determined the shape and stability of the solid–liquid interface [35]. At the interface, the direction of heat transfer was perpendicular to the interface, and the ratio of G/R was high because of the higher G and the lower R. The liquid phase formed large-size columnar dendrites through epitaxial growth on the basis of cellular crystals in the N track deposition. At the left side of the remelting interface, the decrease in temperature gradient led to a decrease in the ratio of G/R. Due to the change in the solid–liquid interface, the liquid phase underwent epitaxial growth to form small-sized dendrites. Finally, the remelting interface with a transition microstructure was formed in this copper composite.

Figure 7a displays the distribution map of each element (Zr, Cu, and Ni) in sample 1 at the middle of the molten pool. It could be observed that the distribution of Zr conformed to the needle-like phase, that the diameter of the needle-like phase was less than 2 μm and that the aspect ratio was approximately 30:1. The Ni dendrites with a diameter of 1–2 μm formed a continuous network structure and connected the needle-like phases together. The distribution maps of Zr, Cu and Ni in sample 2 are shown in Figure 7b. From the SEM image and the distribution map of Zr, it could be clearly observed that the diameter of the needle-like phase was increased to about 4 μm. During the LDED, increasing the laser energy density caused the maximum temperature, size and lifetime of the molten pool to increase, while the cooling rate of the molten pool decreased. Under such conditions, more time and space could be used for the growth of in-situ synthesized needle-like phase, so its size was increased. Furthermore, the content of Ni dendrites decreased, and more independent dendrites and spherical crystals could be observed, resulting in the continuity of the network structure being weakened. There was no continuous network structure formed in sample 2.

Figure 8a shows the TEM image of the needle-like phase, which was combined with Ni or Cu solid solution. A high-resolution analysis was performed on the red dashed box, as shown in Figure 8b. The interface between the needle-like phase and the Ni was clean and without other phases, which facilitated the transfer of load from the matrix to the reinforcement. According to the fast Fourier transform (FFT) for the needle-like phase, the [1$\bar{1}$00] crystal orientation of ZrB_2_ was determined by the calibration of diffraction spots, and the interplanar spacing of the crystal plane (11$\bar{2}$0) was 0.1572 nm. Therefore, it could be determined that the zirconium ceramic needle-like phase was ZrB_2_, which is considered a whisker reinforcement, and its direction of the long axis (preferential growth) was [11$\bar{2}$0].

### 3.4. Growth Mechanism of ZrB_2_ Whisker Reinforcement

The in-situ synthesized ceramic was formed by nucleation and growth in the molten pool. The final shape of this phase was not only related to its own crystal structure, but also the physical and chemical factors of the external matrix. According to the Gibbs–Curie–Wulff theory [36], the equilibrium shape of the crystal should have the lowest total free energy, and so the crystal plane with low surface energy was easily retained in the final shape. Under the ideal condition of equilibrium, ZrB_2_ crystals grew into regular geometric shapes with multiple symmetry planes. The interface structure and surface energy played key roles in the growth of the crystal plane. The interface structure could be divided into a smooth interface and a rough interface according to the crystal structure and bonding bonds between atoms. The Jackson factor (*α*) was used to determine the interface type (*α* > 2, smooth; *α* < 2, rough) [35]:(2)$$\alpha =\frac{\Delta H_{f}}{kT_{m}}\left(\frac{\eta }{v}\right)=\frac{\Delta S_{m}}{R}\left(\frac{\eta }{v}\right)$$
where *k* is Boltzmann’s constant; *T_m_* is the melting point; $\Delta H_{f}$ is the latent heat; *η* is the coordination number between atoms in the layer and the atoms in the same layer (only the surface layer); *v* is the coordination number between the surface layer atoms and the next layer of solid atoms; *R* is the gas constant and $\Delta S_{m}$ is the melting entropy. The space group of ZrB_2_ with a C32 hexagonal close-packed structure was P6/mmm (191). *η*/*v* could be calculated for each crystal plane from the positions of atoms in the hexagonal close-packed structure. The *η*/*v* of {0001}, {10$\bar{1}$0} and {11$\bar{2}$0} crystal plane was 6/12, 2/12 and 2/12, respectively. Furthermore, $\Delta S_{m}$ = 97.08 J/mol·K [37], *R* = 8.31 J/mol·K, and so the *α* of the ZrB_2_ low-index were *α*{0001} = 5.84, *α*{10-10} = 1.95 and *α*{11-20} = 1.95. The α of {10$\bar{1}$0} and {11$\bar{2}$0} crystal plane were less than two, so the growth interface of these two crystal plane groups was considered as a rough interface. The process of atom attachment to the rough surface could be achieved with only a small kinetic driving force. Therefore, the growth rate of {10$\bar{1}$0} and {11$\bar{2}$0} crystal planes was greater than that of the {0001} crystal plane. Furthermore, the crystal planes with smaller interplanar spacing had lower surface energy and a faster growth rate. According to the Bravais–Friedel–Donnay–Harker (BFDH) model [38,39], the growth rate (R_uvtw_) of a crystal plane (uvtw) was inversely proportional to the atomic layer distance (d_uvtw_) of the crystal plane. According to the interplanar spacing of the common low-index crystal plane families of ZrB_2_ (Table 2), the order of the growth rate of the crystal planes was: {11$\bar{2}$0} > {10$\bar{1}$1} > {10$\bar{1}$0} > {0001}. The crystal plane family {11$\bar{2}$0} had the fastest growth rate. During the laser processing, there were temperature gradients and composition gradients due to rapid cooling and the solute flow in the molten pool. Therefore, the preferential growth direction <11$\bar{2}$0> of the ZrB_2_ crystal nucleus was paralleled to the heat transfer direction. As shown in Figure 8b, the long axis direction of needle-shaped ZrB_2_ was [11$\bar{2}$0], which was consistent with the analysis of the BFDH model.

Figure 9a shows the TEM image of needle-like ZrB_2_. The regular crystal planes at the tip of ZrB_2_, which are a typical morphology of the single crystal, could be observed. Furthermore, the angles between the crystal planes were approximately 120°, which is similar to the ideal hexagonal morphology. Figure 9b presents the enlarged view of the tip of ZrB_2_ as marked by the dashed box. Two symmetrical facets could be clearly observed, and the normal angle θ between the two facets was about 62°. The space group of ZrB_2_ with a C32 close-packed hexagonal structure was P6/mmm (191), and the 4-axis coordinate system of its crystal structure is shown in Figure 9c. According to the vector relationship of each crystal orientation in the hexagonal close-packed crystal structure, the long axis direction of [11$\bar{2}$0] and the included angle θ, the crystal orientation of the two facets at the growth tip were determined to be [10$\bar{1}$0] and [01$\bar{1}$0] and the crystal orientation of the side was [1$\bar{1}$00] and [$\bar{1}$100], which belong to the {10$\bar{1}$0} crystal plane family. Therefore, the surface of the ZrB_2_ crystal was mainly composed of {0001} and {10$\bar{1}$0}, which was consistent with the theoretical analysis. As a result, under the combined action of its own crystal structure and non-equilibrium thermodynamic conditions, the ZrB_2_ crystal nucleus grew into a whisker reinforcement.

### 3.5. Microhardness of the Composites

The contour map of the microhardness reflects the spatial homogeneity of the structure and the properties of the composites. Figure 10a presents the distribution of microhardness on the BD-LD surface for sample 1. The highest microhardness was 322.06 HV_0.2_. The average microhardness, 294.26 HV_0.2_, was about five times higher than that of pure copper. The fine ZrB_2_ needle-like phase and Ni dendritic network structure improved the microhardness of sample 1. However, there were fewer needle-like phases and the large-size columnar Ni dendrites at the remelting interface (Figure 4), which was the region of poor reinforcement, so the microhardness was relatively low. As shown in Figure 10b, compared with sample 1, the microhardness distribution of sample 2 was more uniform, but the average hardness was decreased to 275.08 HV_0.2_. As can be seen from Figure 8b, the size of the ZrB_2_ inside sample 2 was larger, and the Ni dendrites did not constitute a continuous network structure, which led to a decrease in the hardness of the composite material.

### 3.6. Tensile Strength of the Composites

The tensile test was carried out at room temperature, and the stress–strain curves of the ZrB_2_ reinforced Cu composites is shown in Figure 11. Table 3 shows the ultimate tensile strength and elongation of samples 1 and 2. Sample 1, which was prepared at a laser energy density of 0.20 kJ/mm^2^, exhibited a higher tensile strength with the maximum value of 535.24 MPa. Its average ultimate tensile strength and elongation was 513.19 MPa and 7.8%, respectively. The average tensile strength and elongation of sample 2 were 368.96 MPa and 10.15%, respectively.

Figure 12 displays the fracture morphology of tensile samples for the ZrB_2_ reinforced Cu composites. It could be observed that there were dimples and planes on the fracture surface of sample 1, so the fracture mechanism of the composite material was a mixed mechanism of ductile and brittle fracturing, as shown in Figure 12a. In the enlarged SEM image (Figure 12b), it could be observed that the shape of the dimples was similar to the morphology of the Ni dendrites. Furthermore, the distance between the tear ridges was about 2 μm, which was close to the spacing of the Ni secondary dendrite. The tearing edge was therefore considered to be formed by the ductile fracture of the Cu matrix between the secondary dendrites. In addition, long pits could be observed at the arrow marks, which were formed by pulling out the needle-like ZrB_2_ during the fracture process. The interface strength between the Cu matrix and Ni dendrites was relatively weak, so the cracks were prone to propagate at this grain boundary, which caused the intergranular fracture. Figure 12c displayed the fracture morphology of tensile sample 2. Only a large number of dimples and a tear ridge similar to sample 1 were observed, with the average size of dimples being about 8 μm. As shown in Figure 12d, the fractured reinforcing phase could be observed on the fracture morphology. For sample 1, ZrB_2_ with high elastic modulus could bear the main load, which is the key to significantly improving the mechanical properties of composites. The network of Ni dendrites connected the ZrB_2_ whiskers together and could effectively transfer external loads from the matrix to the reinforcements, which improved the tensile strength of the composite material. For sample 2, due to the change in the microstructure (the size of ZrB_2_ increased, whereas the number and continuity of Ni dendrites decreased), the tensile strength reduced, but the elongation increased.

## 4. Conclusions

The needle-like ZrB_2_ reinforced Cu composites were successfully fabricated using laser direct energy deposition. The microstructure of the composites was mainly a composite structure composed of needle-like ZrB_2_, Ni dendrites, and a Cu matrix.In-situ synthesized ZrB_2_ was shown to be a high-strength whisker reinforcement. In sample 1, the diameter of ZrB_2_ was less than 2 μm, and the aspect ratio was about 30:1. The long axis direction with the fastest growth rate of ZrB_2_ ceramic whiskers was {11-20}.In the Cu matrix, a network of Ni dendrites connected the needle-like ZrB_2_. The morphological changes of Ni dendrites could be observed at the interface inside the composite material: cellular crystals → large-sized columnar dendrites → small-sized dendrites (along the solidification direction).ZrB_2_ ceramic reinforcement and Ni dendrites significantly improved the mechanical properties of the composite material. The microhardness and ultimate tensile strength of the composite material reached 322 HV_0.2_ and 535 MPa, respectively, and the tensile fracture of sample 1 was a mixed mechanism of ductile and brittle fracturing. As the laser energy density increased, the size of the in-situ synthesized ZrB_2_ increased, and the number and continuity of Ni dendrites decreased, which decreased the hardness and strength of the composite material, and the fracture mechanism of sample 2 was changed to ductile fracturing.

## Figures and Tables

**Figure 1 micromachines-13-00212-f001:**
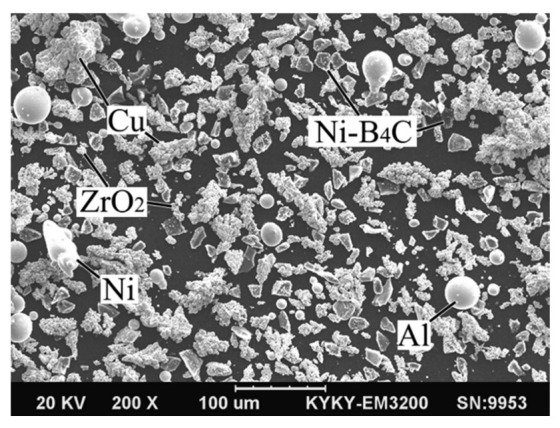
SEM image of the composite mixed powders.

**Figure 2 micromachines-13-00212-f002:**
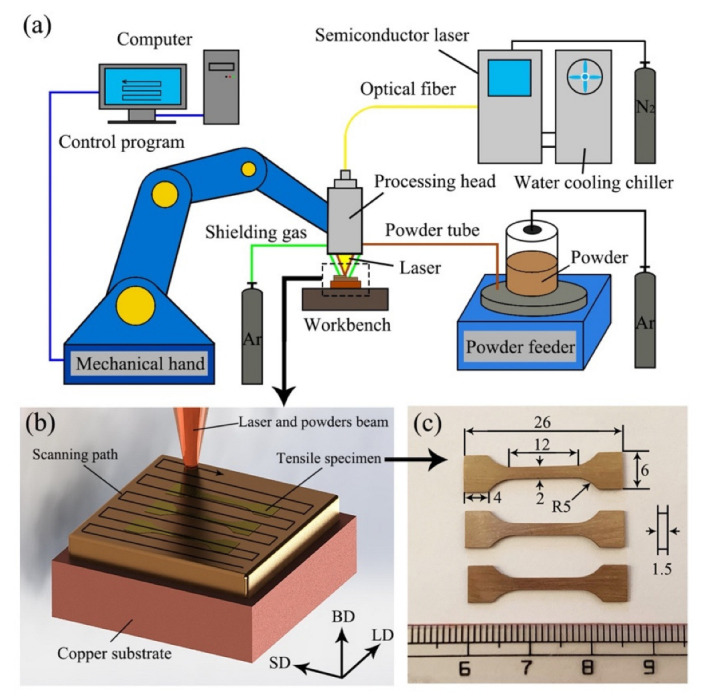
Schematic diagram of (**a**) the laser direct energy deposition system and (**b**) a 3D sample. (**c**) Size of the tension sample.

**Figure 3 micromachines-13-00212-f003:**
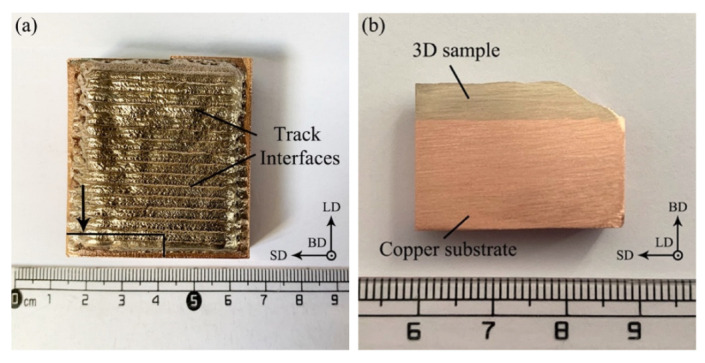
(**a**) Macroscopic surface morphology and (**b**) cross-section image of sample 1.

**Figure 4 micromachines-13-00212-f004:**
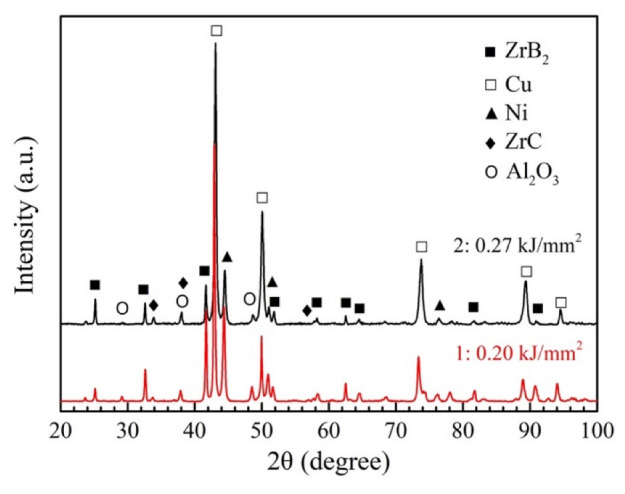
Phase constituents of composite material prepared using laser direct energy deposition at different laser energy densities.

**Figure 5 micromachines-13-00212-f005:**
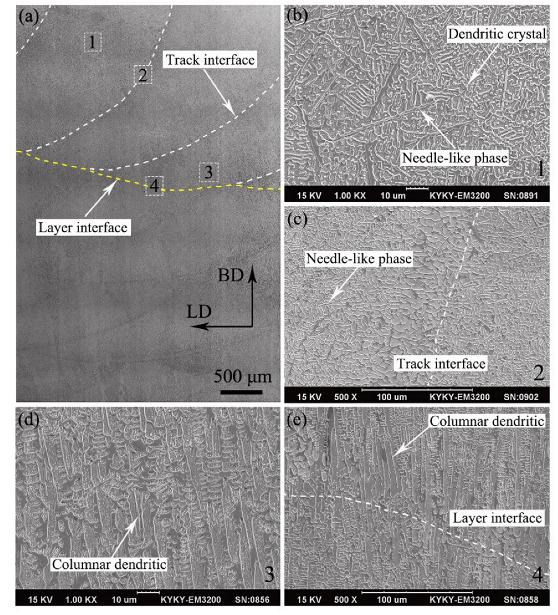
(**a**) SEM image of sample 1 on the BD-LD plane. High magnification SEM images of (**b**) the middle of the molten pool; (**c**) the track interface; (**d**) the bottom of the molten pool; (**e**) the layer interface.

**Figure 6 micromachines-13-00212-f006:**
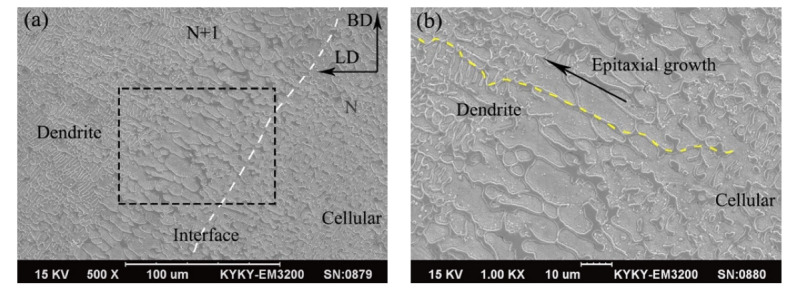
(**a**) Microstructure and (**b**) remelting interface of sample 2 on the BD-LD plane.

**Figure 7 micromachines-13-00212-f007:**
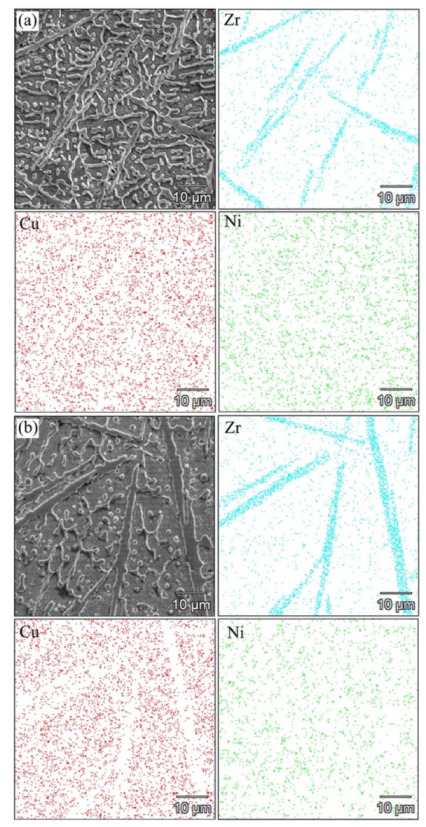
Microstructure and distribution map of elements for (**a**) sample 1 and (**b**) sample 2.

**Figure 8 micromachines-13-00212-f008:**
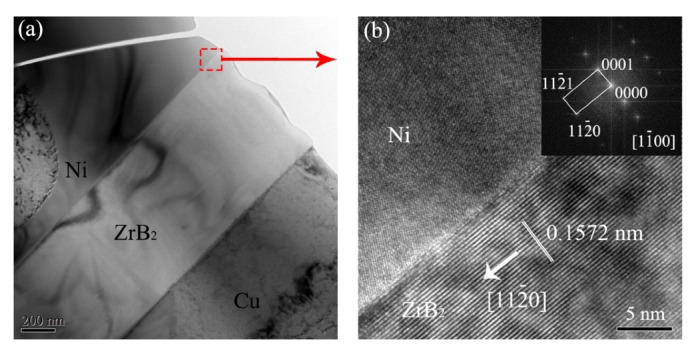
(**a**) TEM image of the needle-like phase. (**b**) HRTEM image of the interface between ZrB_2_ and Ni dendrite.

**Figure 9 micromachines-13-00212-f009:**
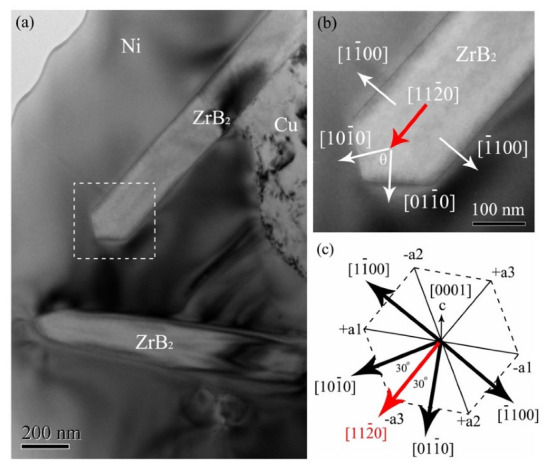
TEM image of (**a**) needle-like ZrB_2_ and (**b**) the growth tip for ZrB_2_. (**c**) The coordinate system of ZrB_2_ crystal structure.

**Figure 10 micromachines-13-00212-f010:**
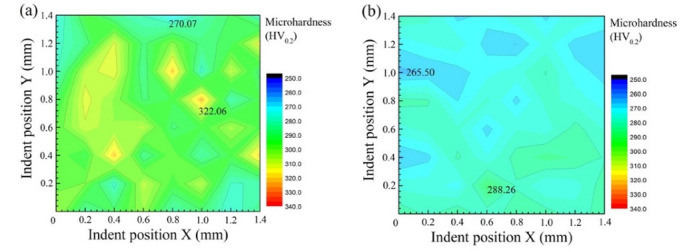
Distribution of microhardness on the BD-LD surface for (**a**) sample 1 and (**b**) sample 2.

**Figure 11 micromachines-13-00212-f011:**
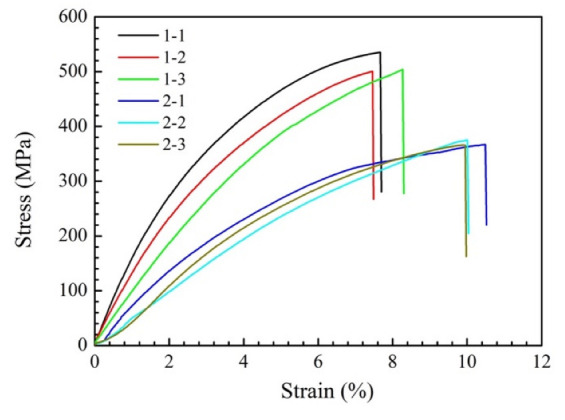
Tensile strength of the ZrB_2_ reinforced Cu composites.

**Figure 12 micromachines-13-00212-f012:**
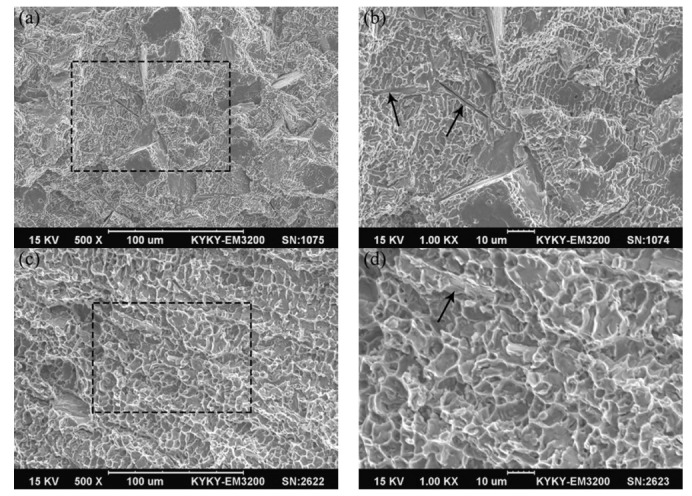
Fracture morphology of tensile samples for the ZrB_2_ reinforced Cu composites: (**a**,**b**) sample 1, (**c**,**d**) sample 2.

**Table 1 micromachines-13-00212-t001:** Purity, size and contents of the composite powders.

Material	Cu	Ni	ZrO_2_	Al	Ni-B_4_C
Purity (%)	99.9	99.9	99.9	99.5	99.5
Size (μm)	53–75	48–75	25–48	38–75	25–48
Content (wt%)	70.00	13.96	10.41	3.04	2.59

**Table 2 micromachines-13-00212-t002:** The crystal plane families and interplanar spacing of ZrB_2_.

{uvtw}	{0001}	$\left\{10\bar{1}0\right\}$	$\left\{10\bar{1}1\right\}$	$\left\{11\bar{2}0\right\}$
d_{uvtw}_ (Å)	3.5305	2.7445	2.1663	1.5843

**Table 3 micromachines-13-00212-t003:** Ultimate tensile strength and tensile elongation of the samples 1 and 2.

Number	Ultimate Tensile Strength (MPa)	Tensile Elongation (%)	Mean Ultimate Tensile Strength (MPa)	Mean Tensile Elongation (%)
1-1	535.24	7.67	513.19	7.80
1-2	500.46	7.46		
1-3	503.88	8.27		
2-1	366.77	10.49	368.96	10.15
2-2	374.85	10.01		
2-3	365.27	9.95		

## Data Availability

Data sharing is not applicable to this article.

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
