# Peer review of "Microstructure Evolution and Mechanical Properties of Needle-like ZrB2 Reinforced Cu Composites Manufactured by Laser Direct Energy Deposition"

_micromachines, 2022, doi:10.3390/mi13020212_

Round 1
Reviewer 1 Report
This is an interesting work and can be accepted for publication after some improvements:
1
In the title and the whole paper, authors emphasize “Needle-like ZrB2 Reinforced Cu Composites”, but from the authors’ work, it can be seen that in addition to ZrB2, the amount and the distribution of Ni also have a very important influence on the properties of samples. It might be more important than ZrB2. How to identify their influences respectively? Which one is more important? It could be better if authors can give some explanations.
2
In page 9, authors mentioned “smooth” and “rough” interface types. Please give a more clear definition: what are “smooth” and “rough”? They are related to interfacial energy or others?
3
In line 273 of page 9, υ should be ν
4
In page 9, authors mentioned “ZrB2, the η/v of {0001}, {10-10}, and {11-20} crystal plane was 6/12, 2/12, and 2/12, respectively”. They should give explanation of how to get these numbers.
5
In page 9, authors should give reference for “S = 97.08 J/mol·K”
6
In page 9, authors mentioned “The crystal planes with larger interplanar spacing had lower surface energy and a faster growth rate. According to the Bravais-Friedel- Donnay-Harker (BFDH) model [35, 36], the growth rate Ruvtw of a crystal plane (uvtw) was inversely proportional to the atomic layer distance duvtw of the crystal plane.”
These descriptions are confusing and contradictory.
Also, what are “Ruvtw” and “duvtw”?
7
In page 9, “the crystal orientation parallel to the heat transfer direction in the <11-20> crystal orientation family would be the preferential growth direction of the ZrB2 crystal nucleus.” This description is confusing.
8
10-10 should be written to be 10Ī0. In the whole paper, there are many of the same writing issues.
9
Authors observed needle-like ZrB2 in Ni/Cu matrix. Usually, needle-like precipitates have a crystallographic orientation relationship with the matrix. In the authors’ work, it would be interesting to know if authors observed such a relationship.
Author Response
Reviewer 1:
This is an interesting work and can be accepted for publication after some improvements:
- In the title and the whole paper, authors emphasize “Needle-like ZrB2 Reinforced Cu Composites”, but from the authors’ work, it can be seen that in addition to ZrB2, the amount and the distribution of Ni also have a very important influence on the properties of samples. It might be more important than ZrB2. How to identify their influences respectively? Which one is more important? It could be better if authors can give some explanations.
Response 1: Thanks for the reviewer’s comments. The physical properties of ZrB2 were better than those of Ni. The melting point of Ni is 1455 oC, the elastic modulus is 207 GPa, and the microhardness is 350 HV. While the melting point of ZrB2 is 3245oC, the elastic modulus is 489 GPa, and the microhardness is 23 GPa (~2355HV). In this paper, Ni could improve the solid solubility of the solute, so that the ceramic phase (solute) could be better combined with the matrix, thereby improving the bonding strength of the ceramic reinforcement and the matrix after solidification. And Ni could transfer the load to the ceramic reinforcement. The in-situ synthesized ZrB2 was a single crystal with physical properties close to the theoretical values, which had excellent reinforcement efficiency and could bear the main load. Therefore, ZrB2 was more important that Ni in the composites.
We added the explanation on page 12. “ZrB2 with high elastic modulus could bear the main load, which was the key to significantly improve the mechanical properties of composites. The network of Ni dendrites connects the ZrB2 whiskers together and could effectively transfer external loads from the matrix to the reinforcements, which improved the tensile strength of the composite material.”
- In page 9, authors mentioned “smooth” and “rough” interface types. Please give a more clear definition: what are “smooth” and “rough”? They are related to interfacial energy or others?
Response 2: “Smooth” and “rough” were definition that characterize the microstructure of the solid-liquid interface.
Rough interface: On the solid phase side of the interface, the solid phase atoms occupied about 50% of the lattice positions, forming an uneven discontinuous interface structure.
Smooth interface: On the solid phase side of the interface, the solid phase atoms occupied almost all lattice positions, with only a few vacancies and steps, forming a smooth and flat interface structure.
The rough interface had higher surface energy. The process of atom attachment to the rough surface could be achieved with only a small kinetic driving force. Therefore, the growth rate of rough interfaces was relatively fast.
- In line 273 of page 9, υ should be ν
Response 3: Thanks for the reviewer’s comments. the υ was be changed to ν.
- In page 9, authors mentioned “ZrB2, the η/v of {0001}, {10-10}, and {11-20} crystal plane was 6/12, 2/12, and 2/12, respectively”. They should give explanation of how to get these numbers.
Response 4: The crystal structure of ZrB2 was C32 hexagonal close-packed structure (P6/mmm, 191). From the arrangement positions of atoms in the hexagonal close-packed structure, η/v could be calculated for each crystal plane.
We added the explanation on page 9.
- In page 9, authors should give reference for “S = 97.08 J/mol·K”
Response 5: Thanks for the reviewer’s comments. We added the reference [37] for “S = 97.08 J/mol·K”
- In page 9, authors mentioned “The crystal planes with larger interplanar spacing had lower surface energy and a faster growth rate. According to the Bravais-Friedel- Donnay-Harker (BFDH) model [35, 36], the growth rate Ruvtw of a crystal plane (uvtw) was inversely proportional to the atomic layer distance duvtw of the crystal plane.”
These descriptions are confusing and contradictory.
Also, what are “Ruvtw” and “duvtw”?
Response 6: Thanks for the reviewer’s comments. We made a mistake in this sentence. The sentence was changed to “The crystal planes with smaller interplanar spacing had lower surface energy and a faster growth rate.”.
Ruvtw and duvtw were the growth rate and the interplanar spacing of the crystal plane with the crystal plane index (uvtw), respectively. For clarity of description, this sentence had been modified to “According to the Bravais-Friedel-Donnay-Harker (BFDH) model [38, 39], the growth rate (Ruvtw) of a crystal plane (uvtw) was inversely proportional to the atomic layer distance (duvtw) of the crystal plane.”.
- In page 9, “the crystal orientation parallel to the heat transfer direction in the <11-20> crystal orientation family would be the preferential growth direction of the ZrB2 crystal nucleus.” This description is confusing.
Response 7: Thanks for the reviewer’s comments. For clarity of description, this sentence was changed to “ZrB2 grew into needle-like shape along the heat flow direction, and its long axis direction (fast growth direction) was <11-20> crystal orientation family.”.
- 10-10 should be written to be 10Ī0. In the whole paper, there are many of the same writing issues.
Response 8: We made changes based on reviewer’s comments.
- Authors observed needle-like ZrB2 in Ni/Cu matrix. Usually, needle-like precipitates have a crystallographic orientation relationship with the matrix. In the authors’ work, it would be interesting to know if authors observed such a relationship.
Response 9: Thanks for the reviewer’s comments. In this work, ZrB2 was in-situ synthesized in the melt, which was different with diffusion-type precipitates. There was no special crystal orientation relationship between the needle-like precipitate and the matrix.
Reviewer 2 Report
In this research, the authors fabricated needle-like ZrB2-reinforced Cu composites using the laser direct energy deposition process. The research’s focus is the in-situ formation of reinforcement particles. Microstructure evolution and mechanical properties are also studied. Following are the limitations that should be addressed.
- There are grammatical errors. English needs to be polished.
- The Introduction didn’t include the literature review of ZrB2-reinforced Cu composites. The novelty of this work is not clear.
- In the Materials and Method section, it is better to include one SEM micrograph to characterize the morphology of the feedstock powders.
- For the blown-powder AM process, particularly when the premixed powders are used, the actual composition will deviate from the premixed composition. Please do EDS to check this deviation.
- The authors only tested two energy densities. How to determine the optimal condition?
- There are many variables that will affect the final properties, such as the laser deposition parameters, the particle size and shape of the premixed powders, the grain size and shape of the resolidified reinforcements. The influences of these variables are not well controlled or studied.
- This manuscript lacks a thorough discussion of the results.
Author Response
Reviewer 2:
In this research, the authors fabricated needle-like ZrB2-reinforced Cu composites using the laser direct energy deposition process. The research’s focus is the in-situ formation of reinforcement particles. Microstructure evolution and mechanical properties are also studied. Following are the limitations that should be addressed.
- There are grammatical errors. English needs to be polished.
Response 1: Thanks for the reviewer’s comments. We have checked, reviewed and corrected the grammar of the paper.
- The Introduction didn’t include the literature review of ZrB2-reinforced Cu composites. The novelty of this work is not clear.
Response 2: Thanks for the reviewer’s comments. We added related literature [23, 31] and introductions of ZrB2-reinforced Cu composites, see page 2 for details.
- In the Materials and Method section, it is better to include one SEM micrograph to characterize the morphology of the feedstock powders.
Response 3: Thanks for the reviewer’s comments. We added the SEM micrograph of the composite mixed powders (Figure 1).
- For the blown-powder AM process, particularly when the premixed powders are used, the actual composition will deviate from the premixed composition. Please do EDS to check this deviation.
Response 4: Thanks for the reviewer’s comments. We analyzed the deviation of raw powders with different reactant contents (including the premixed powders in this work) from the actual composition through previous experiments. According to the results of EDS and XRD, the in-situ synthesis reaction was complete, and the actual phases were the same as the target products, indicating that the ratio between the reactants did not change. And no other phases were formed and no reactants remained. The shape of Cu powder was irregular (Figure 1), and its laser energy absorption rate was low, resulting in a lower utilization rate of Cu powder than other powders. Under the process parameters of this work, compared with the premixed composition, the composition deviation of Cu (not participating in the synthesis reaction) was about 5.3%, and the deviation of other components was less than 2%. Therefore, this deviation did not affect the in-situ synthesis reaction. And the actual composition of the composite material could be considered to be the same as the premixed composition.
- The authors only tested two energy densities. How to determine the optimal condition?
Response 5: Thanks for the reviewer’s comments. We prepared Cu composites with different reinforcement contents (5-40 wt%) under different laser power (1400-2800 W) and scanning speed (1-6 mm/s). Macroscopic morphology, microstructure and mechanical properties of the composites could be affected by the process parameters and the reinforcement content. By comparing the experimental results, the optimal process parameters of the composites with different reinforcement contents were determined. For the Cu composite of this work, the laser power of 1800 W and scanning speed of 3 mm/s (the laser energy density of 0.20 kJ/mm2) were the optimal conditions, namely the process conditions of sample 1 were the optimal condition, its maximum tensile strength was 535 MPa, and the microhardness reached 322 HV0.2. When the laser energy density was lower than the optimum, due to insufficient energy the macroscopic defects (pores and impurity) gradually increased and the content of in-situ synthesized reinforcement decreased resulting in the decrease of tensile strength and microhardness. For example, at the 0.16 kJ/mm2 of laser energy density, the tensile strength of the composites was 417 MPa, and the microhardness was 284 HV0.2. When the laser energy density was higher than the optimum, the grain size increased due to the increase of the molten pool lifetime and the decrease of the cooling rate, resulting in the decrease of the mechanical properties. Such as sample 2.
In this work, we mainly analyzed the microstructure and interface of copper matrix composites prepared by laser direct energy deposition, the in-situ synthesis and morphology of the reinforcements. The effect of increasing the laser energy density on the microstructure and reinforcement was analyzed, the mechanical properties of the composites were tested, and the reinforcement mechanism of the reinforcement was discussed. Limited by the length of the article and many factors affecting the optimal process conditions, we would introduce the effect of process parameters (including combinations of different laser powers and scanning speeds under the same laser energy density) on composites with different reinforcement contents in the subsequent work. Such as macroscopic morphology, microstructure, distribution and morphology of reinforcements, and mechanical properties. The optimum combination of reinforcement content, properties and process parameters of the composite was determined by comprehensive analysis of the experimental results.
- There are many variables that will affect the final properties, such as the laser deposition parameters, the particle size and shape of the premixed powders, the grain size and shape of the resolidified reinforcements. The influences of these variables are not well controlled or studied.
Response 6: Thanks for the reviewer’s comments. We agree that there were many variables that will affect the final properties of the Cu composites.
We use the same raw powder, so the particle size and shape of the premixed powder were the same.
Both the grain size and shape of the resolidified structure were closely related to the laser deposition parameters. Therefore, under the condition of only controlling the laser energy density and other variables being the same, we studied the change of the size and shape of the reinforcement, and discussed the relationship between the change of the microstructure and the mechanical properties further.
- This manuscript lacks a thorough discussion of the results.
Response 7: Thanks for the reviewer’s comments. We have added the discussion of the results, detailed in the manuscript.
Reviewer 3 Report
The authors investigated the fabrication of metal matrix composite using laser direct energy deposition. They successfully synthesized in-situ ZrB2 ceramic reinforcement in the Cu matrix. They characterized the samples carefully using TEM, SEM, and mechanical properties of the fabricated parts were tested. Therefore, I recommend the manuscript for publication in its current state.
Author Response
Reviewer 3:
The authors investigated the fabrication of metal matrix composite using laser direct energy deposition. They successfully synthesized in-situ ZrB2 ceramic reinforcement in the Cu matrix. They characterized the samples carefully using TEM, SEM, and mechanical properties of the fabricated parts were tested. Therefore, I recommend the manuscript for publication in its current state.
Response: Thanks for the reviewer’s comments.
Round 2
Reviewer 1 Report
This manuscript has been improved and can be accepted for publication